# Giant magnetocaloric effect in a rare-earth-free layered coordination polymer at liquid hydrogen temperatures

J. J. B. Levinsky [1,2], B. Beckmann [3], T. Gottschall [4], D. Koch[5], M. Ahmadi [1], O. Gutfleisch[3] & G. R. Blake [1]✉

Magnetic refrigeration, which utilizes the magnetocaloric effect, can provide a viable alternative to the ubiquitous vapor compression or Joule-Thompson expansion methods of refrigeration. For applications such as hydrogen gas liquefaction, the development of magnetocaloric materials that perform well in moderate magnetic fields without using rare-earth elements is highly desirable. Here we present a thorough investigation of the structural and magnetocaloric properties of a novel layered organic-inorganic hybrid coordination polymer $Co_4(OH)_6(SO_4)_2[enH_2]$ ($enH_2$ = ethylenediammonium). Heat capacity, magnetometry and direct adiabatic temperature change measurements using pulsed magnetic fields reveal a field-dependent ferromagnetic second-order phase transition at 10 K < $T_C$ < 15 K. Near the hydrogen liquefaction temperature and in a magnetic field change of 1 T, a large maximum value of the magnetic entropy change, $\Delta S_M^{Pk} = -6.31\,J\,kg^{-1}\,K^{-1}$, and an adiabatic temperature change, $\Delta T_{ad} = 1.98\,K$, are observed. These values are exceptional for rare-earth-free materials and competitive with many rare-earth-containing alloys that have been proposed for magnetic cooling around the hydrogen liquefaction range.

The magnetocaloric effect, which is the temperature change of a material under an applied magnetic field, has the potential to form the basis of an environmentally friendly alternative to currently ubiquitous vapor compression refrigeration methods. For room temperature cooling applications based on the vapor compression method of cooling, magnetic refrigeration promises larger thermodynamic efficiencies and does not rely on the use of volatile refrigerants with large global warming potentials[1–4]. Similar efficiency improvements are promised for cryogenic temperature cooling applications based on Joule-Thompson expansion such as hydrogen gas liquefaction ($T_{boil}$ = 20.3 K at atmospheric pressure)[5].

In the context of growing interest on the industrial level for using hydrogen on a large scale as an alternative energy source towards carbon neutrality, reducing the associated energy and economic cost of hydrogen liquefaction is crucial. Towards this end magnetic refrigeration can play an important role. In particular, it is important to develop magnetocaloric materials comprised of earth-abundant elements and with excellent performance under the moderate magnetic field changes achievable with permanent magnets ($\Delta\mu_o H \leq 2\,T$) instead of more expensive superconducting magnets.

The magnetocaloric effect is characterized by the adiabatic temperature change, $\Delta T_{ad}$, and isothermal entropy change, $\Delta S_T$, of a material under an applied magnetic field. Large magnetic moments,

[1]Zernike Institute for Advanced Materials, University of Groningen, Nijenborgh 3, 9747AG Groningen, The Netherlands. [2]EaStCHEM School of Chemistry and Centre for Science at Extreme Conditions, University of Edinburgh, Joseph Black building, David Brewster Road, EH9 3FJ Edinburgh, UK. [3]Functional Materials, Institute of Materials Science, Technical University of Darmstadt, Darmstadt 64287, Germany. [4]Dresden High Magnetic Field Laboratory (HLD-EMFL), Helmholtz-Zentrum Dresden-Rossendorf (HZDR), Dresden 01328, Germany. [5]Structure Research, Institute of Materials Science, Technical University of Darmstadt, Darmstadt 64287, Germany. ✉e-mail: g.r.blake@rug.nl

typically associated with rare-earth elements, maximize the potential magnetic entropy change, $\Delta S_M$, under an applied field[6–8] as exemplified by the prototypical room temperature magnetocaloric material $Gd_5Si_2Ge_2$[9]. Therefore, the search for new and better performing magnetocaloric materials at cryogenic temperatures has centered mostly around exploring rare-earth containing ferromagnetic or metamagnetic intermetallics such as the $RT_2$ Laves phases, $RTX$ and $RT_2X_2$ families of compounds where $R$ is a rare-earth element, $T$ is a transition metal element and $X$ is a p-block element[8,10–17].

A different approach to optimize magnetocaloric properties can be found in the class of organic-inorganic hybrid materials in which the control of structure-property relationships is exerted by selection of their (in)organic building blocks. Thus far, this approach has been most successful in the ultra-low temperature range with $T < 2$ K, in the form of Gd-based molecular clusters and coordination polymers[17–20] in which magnetic long-range order (LRO) is typically suppressed due to the large separation between the metal sites. Materials such as $Gd(HCOO)_3$ and $Gd(OH)CO_3$, with one- and two-dimensional (2D) structures, respectively, show comparable or better magnetocaloric properties than the prototypical three-dimensional ultra-low-temperature magnetocaloric $Gd_3Ga_5O_{12}$ with reported $\Delta S_M > 50$ J kg$^{-1}$ K$^{-1}$ for $\Delta\mu_o H = 7$ T[21,22]. However, due to the general scarcity and resource criticality of rare-earth elements, there is growing interest in finding transition-metal-based alternatives[23–26]. In the hydrogen liquefaction temperature regime, it has previously been shown that the layered materials β-$Co(OH)_2$ and $CrCl_3$ are competitive with rare-earth-based magnetocaloric compounds. Long-range magnetic order sets in for both compounds between 10 and 20 K, resulting in both cases in a large $\Delta S_M$ of ~15 J kg$^{-1}$ K$^{-1}$ for $\Delta\mu_o H = 5$ T in the 15 − 20 K range[27–29]. $Co_3V_2O_8$, which can be considered as a layered material with respect to the kagomé staircase lattice formed by the magnetic ions, exhibits an even higher $\Delta S_M$ of 17 J kg$^{-1}$ K$^{-1}$ for $\Delta\mu_o H = 5$ T at 11 K[30]. $Co_2(OH)_{4-x}Cl_x$ is another compound in which the magnetic ions form planes, in this case of both kagomé and triangular character, and shows a $\Delta S_M$ of 8 J kg$^{-1}$ K$^{-1}$ in a low applied field of $\Delta\mu_o H = 2$ T at 10 K[31]. These studies suggest that layered and anisotropic rare-earth-free materials have great potential for low-temperature magnetocaloric applications.

Here, we report on the structure and magnetocaloric properties of a novel layered brucite-type hybrid material $Co_4(OH)_6(SO_4)_2[enH_2]$ ($enH_2$ = ethylenediammonium). This compound exhibits a ferromagnetic second order phase transition at 10 K $< T_c < 15$ K, with negligible hysteresis, which is highly beneficial for cryogenic magnetocaloric applications[32]. Indirect and direct measurements of the magnetocaloric properties show that in low fields of 1 − 2 T that are easily accessible using permanent magnets, the peak values of $\Delta S_M$ and $\Delta T_{ad}$ for $Co_4(OH)_6(SO_4)_2[enH_2]$ are exceptional for rare-earth-free materials and competitive with many proposed rare-earth-based magnetocalorics in the 15 − 25 K range, relevant for hydrogen liquefaction.

## Results
### Structural investigations
Single crystals of $Co_4(OH)_6(SO_4)_2[enH_2]$, depicted in Fig. 1a, were obtained by solvothermal synthesis. These pink crystals have typical dimensions of roughly 0.5 ×0.5 ×0.2 mm$^3$. Single crystal X-ray diffraction data collected at 107 K were used to elucidate the structure. Crystallizing in the triclinic space group $P\bar{1}$, the structure of $Co_4(OH)_6(SO_4)_2[enH_2]$ is shown in Fig. 1b and c viewed along the $c$- and $a$-axis respectively. A brucite-type ($Mg(OH)_2$) structure is identified, consisting of layers of distorted edge-sharing $Co(OH)_4(SO_4)_2$ and $Co(OH)_5(SO_4)$ octahedra with elongated axial bonds which form a

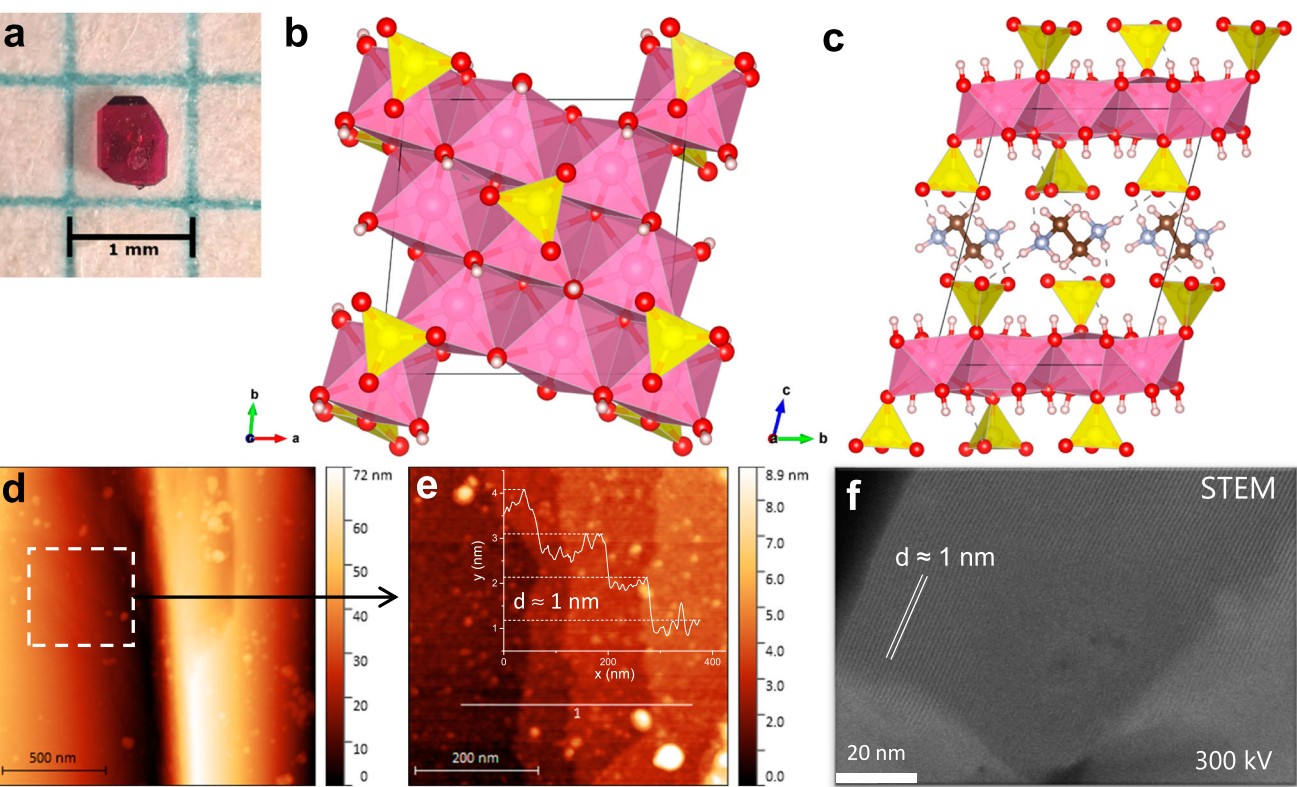

**Fig. 1 | Structure of $Co_4(OH)_6(SO_4)_2[enH_2]$.** **a** Photograph of a single crystal of $Co_4(OH)_6(SO_4)_2[enH_2]$ through an optical microscope. **b, c** Crystal structure of $Co_4(OH)_6(SO_4)_2[enH_2]$ viewed along the $c$-axis and $a$-axis, respectively. Cobalt hydroxide layers are shaded pink, sulfate groups are shaded yellow, oxygen atoms are red, carbon atoms are brown, nitrogen atoms are light blue, and hydrogen atoms are white. **d** AFM topography image of exfoliated nanosheet showing a large step edge. **e** Topography image of the region highlighted by the dashed lines in (**d**). The trace of a line scan is indicated by the white line labeled as 1. In the inset the height is plotted against position along the line scan. **f** STEM image of $Co_4(OH)_6(SO_4)_2[enH_2]$ showing lattice fringes spaced roughly 1 nm apart.

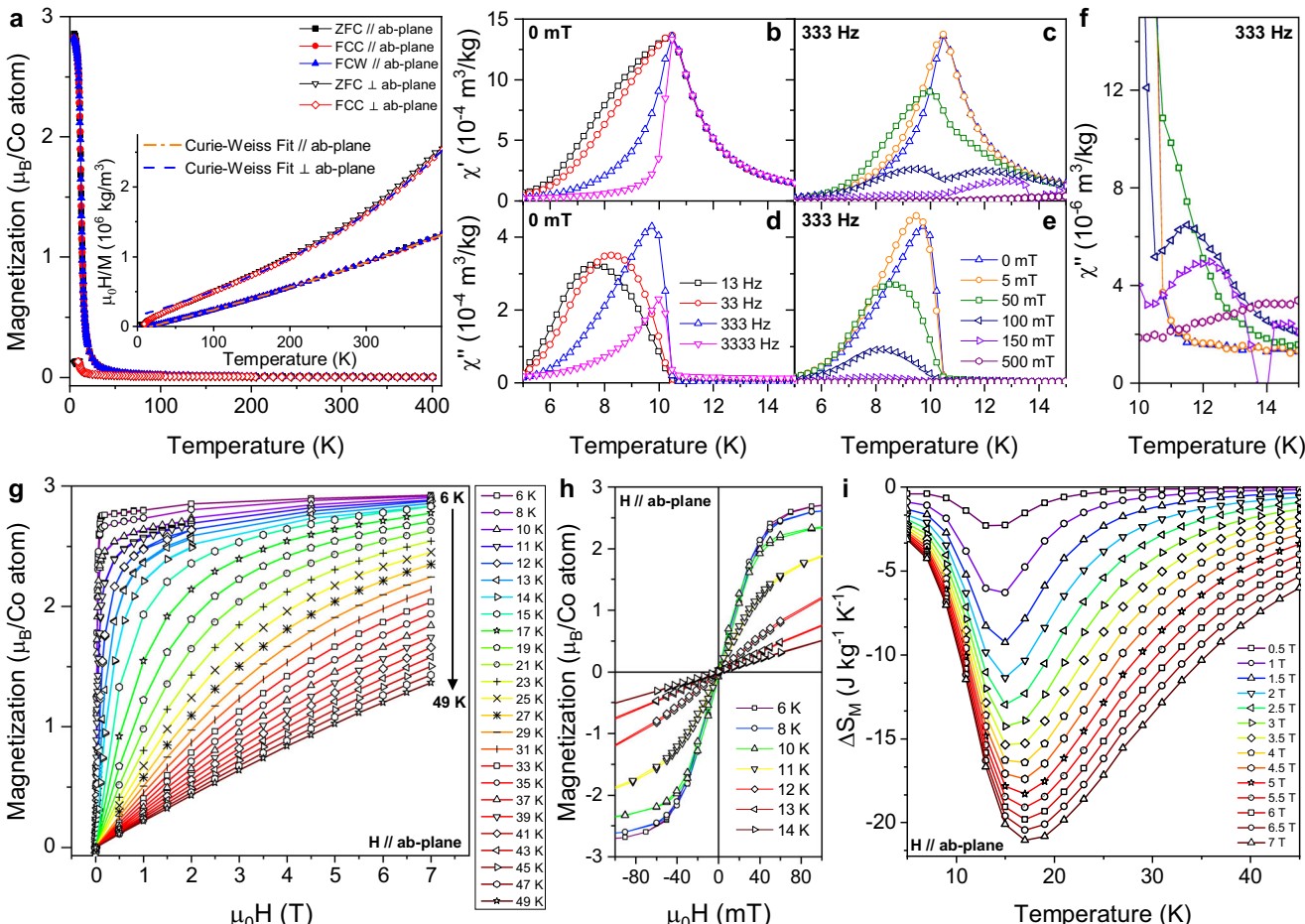

**Fig. 2 | Magnetic properties of Co$_4$(OH)$_6$(SO$_4$)$_2$[enH$_2$]. a** Zero-field-cooled (ZFC), field-cooled-cooling (FCC) and field-cooled-warming (FCW) temperature dependent magnetization measured parallel and perpendicular to the ab-plane of a single crystalline sample in a field of $\mu_o H = 200$ mT. The inset shows the inverse field-normalized magnetization ($\mu_0 H/M$) versus temperature. The dashed lines indicate a Curie-Weiss fit for $T > 300$ K. **b, c** Real and (**d, e**) imaginary components of the temperature-dependent AC magnetic susceptibility. For (**b**) and (**d**) the measurements are performed in the absence of a DC bias field with a superimposed AC field of 0.43 mT oscillating at the four frequencies depicted in the legend. For (**c**) and (**e**) the legend indicates the strength of the applied DC bias field present during the measurements alongside a superimposed AC field of 0.43 mT oscillating at 333 Hz.

Note that the 0 mT data acquired at 333 Hz depicted in both (**b**), (**c**) and (**d**), (**e**) are indicated using the same symbol and color (blue triangles). **f** Close-up of the imaginary part of the AC magnetic susceptibility between 10 and 15 K measured with an AC frequency of 333 Hz where the legend in (**e**) indicates the strength of the DC bias field. Note the difference in the scale used between (**d**), (**e**) and (**f**). **g** Magnetic field dependent magnetization measured isothermally from 6 to 49 K parallel to the ab-plane. **h** Close-up view of magnetization versus magnetic field loops collected at selected temperatures parallel to the ab-plane, highlighting the negligible coercive field. **i** Extracted magnetic entropy change plotted against temperature for magnetic field changes in the range of 0.5 to 7 T.

triangular lattice. The layers are separated by hydrogen-bonded ethylenediammonium (enH$_2$) cations. The presence of hydroxyl groups was confirmed by Fourier transform infrared spectroscopy (Supplementary Information Fig. S1). The octahedral Co-O distances range from 2.028(2) to 2.338(2) Å and are consistent with typical Co$^{2+}$-O distances found in for example CoO and Co(SO$_4$)$_2$(H$_2$O)$_4$[enH$_2$] (2.12 Å and 2.066(1) – 2.112(1) Å respectively)[33,34]. Details of the crystal structure of Co$_4$(OH)$_6$(SO$_4$)$_2$[enH$_2$] and its determination can be found in Section 2 of the Supplementary Information.

Co$_4$(OH)$_6$(SO$_4$)$_2$[enH$_2$] nanosheets can be obtained by sonication-assisted liquid-phase exfoliation in 96 % ethanol and were characterized by atomic force microscopy (AFM) and scanning transmission electron microscopy (STEM). Figure 1d shows an AFM topography image of a step edge, on which a region with terraces can be identified. The step height of the terraces, illustrated by the line scan in Fig. 1e, is roughly 1 nm and corresponds well to the determined interlayer spacing of 10.5382(12) Å. The corresponding phase images to Fig. 1d, e can be found in the Supplementary Information (Fig. S2). In the high-resolution STEM image, shown in Fig. 1f, lattice fringes spaced 1 nm

apart can be resolved, which also corresponds well to the determined interlayer spacing.

## Magnetometry

The temperature dependence of the magnetization, $M(T)$, measured in an applied field of $\mu_o H = 200$ mT parallel and perpendicular to the ab-plane of a single crystal, $M_{\parallel}(T)$ and $M_{\perp}(T)$ respectively, is shown in Fig. 2a. The data were collected using zero-field cooled (ZFC), field-cooled cooling (FCC) and field-cooled warming (FCW) protocols.

$M_{\parallel}(T)$ increases sharply below 20 K and nearly saturates with $M_{\text{sat}} \approx 2.85\,\mu_B$/Co atom at 5 K. $M_{\perp}(T)$ also increases sharply at low temperatures but saturates at a smaller value of $M_{\text{sat}} \approx 0.13\,\mu_B$/Co atom, consistent with strong easy plane anisotropy. No thermal hysteresis is observed in the FCC/FCW curves, although a small deviation between the ZFC and FCC/FCW curves ($M_{\text{ZFC}_{\parallel}}/M_{\text{FC}_{\parallel}} = 0.986$) is visible below $T \approx 11$ K, implying that the magnetic order has a dynamic character. This behavior is consistent with a ferromagnetic phase transition and the magnitude of $M_{\text{sat}}$ implies a high-spin Co$^{2+}$ ($S = 3/2$) ground state.

The temperature dependence of $\mu_0 H/M$, where $M$ is the magnetization measured in $\mu_0 H = 200$ mT, is shown in the inset of Fig. 2a. Notably, deviations from Curie-Weiss behavior are observed from $T_c$ up to 400 K. The high temperature data (100 – 400 K) measured parallel and perpendicular to the $ab$-plane were therefore fitted to the modified Curie-Weiss law ($\chi = \chi_0 + \frac{C}{T - \theta_{cw}}$), resulting in Weiss temperatures of $\theta_\parallel = 21.2 \pm 1.1$ K and $\theta_\perp = -50.1 \pm 1.6$ K respectively. These values indicate the presence of ferromagnetic interactions in the $ab$-plane and antiferromagnetic interactions perpendicular to the $ab$-plane. The temperature independent contributions to the susceptibility are determined to be $\chi_{0\parallel} = 1.59(6) \times 10^{-7}$ m³/kg and $\chi_{0\perp} = 3.89(8) \times 10^{-7}$ m³/kg. The derived effective magnetic moments, $\mu_{eff\parallel}$ and $\mu_{eff\perp}$ are determined to be $2.86 \pm 0.02$ and $2.88 \pm 0.02$ $\mu_B$/Co atom respectively. These values are significantly lower than the spin-only values expected for high-spin Co²⁺ ($S = 3/2$) where $\mu_{eff} = 3.87$ $\mu_B$/Co atom, but are higher than the expected $\mu_{eff}$ of 1.73 $\mu_B$/Co atom for low-spin Co²⁺ ($S = 1/2$). The discrepancy between $\mu_{eff}$ in the 100–400 K range and $M_{sat}$ measured below 20 K might imply the existence of a spin-state transition with temperature or applied magnetic field, for which an associated structural change would be expected due to the Jahn-Teller effect. However, powder x-ray diffraction (PXRD) measurements showed that no significant structural changes occur either on cooling from 300 K to 15 K or at 15 K under applied fields up to 5 T (Supplementary Information Figs. S3-S7), providing strong evidence for the $S = 3/2$ ground state. The anomalous $\mu_{eff}$ values above $T_c$ likely stem from the deviation from Curie-Weiss behavior at high temperatures and may be a result of the lowered spin and lattice dimensionality, or of the presence of short-range correlations similar to Griffiths-like phases[35].

A closer inspection of the magnetic ground state by AC magnetic susceptibility measurements parallel to the $ab$-plane reveals that the ferromagnetic ordering process, as initially suggested by the observed difference in $M_{ZFC}$ and $M_{FC}$, is dynamic in nature. The real ($\chi'(T)$) and imaginary ($\chi''(T)$) parts of the temperature, frequency and magnetic field dependent AC magnetic susceptibility, associated with reversible and irreversible magnetic processes respectively[36,37], are shown in Fig. 2b–f. In the absence of a DC bias field (Fig. 2b), $\chi'(T)$ shows a single peak at 10.5 K, coinciding with the onset of a large peak in $\chi''(T)$ ($\chi''(T)/\chi'(T) = 0.31$) visible in Fig. 2d. Below 10.5 K, $\chi'(T)$ and $\chi''(T)$ abruptly become frequency dependent. With increasing AC frequency the peak in $\chi'(T)$ narrows significantly, whereas the peak in $\chi''(T)$ decreases in magnitude and shifts towards higher temperature. With increasing DC bias field (Fig. 2c, e) the peak in $\chi'(T)$ splits; the higher temperature peak, highlighted in Fig. 2f, emerges at around 11 K when $\mu_0 H \geq 50$ mT and shifts significantly towards higher temperatures with increasing field. In contrast, the frequency-dependent peaks in $\chi'(T)$ and $\chi''(T)$ below 10.5 K both shift towards lower temperature with increasing field, decreasing in magnitude and becoming too weak to resolve when $\mu_0 H \geq 500$ mT.

The $M(\mu_0 H)$ curve measured along the $ab$-plane, $M_\parallel(\mu_0 H)$, depicted in Fig. 2g, shows that for $T < T_c$ in small fields of $\mu_0 H < 1$ T, the magnetization nearly saturates with $M_{sat} \approx 2.80 \mu_B$/Co atom. Far above $T_c$, at 49 K, $M_\parallel(\mu_0 H)$ still reaches values of 1.37 $\mu_B$/Co atom for $\mu_0 H = 7$ T. In Fig. 2h the formation of a narrow hysteresis loop with a coercive field $\mu_0 H_{coercive} \approx 5$ mT is observed below 12 K. This is an order of magnitude smaller than the coercive field observed in the similar ferromagnetic layered compound $Co_5(OH)_6(SO_4)_2(H_2O)_4$[38].

Through Maxwell's relations, $\Delta S_M$ can be indirectly obtained as follows:

$$\Delta S_M = \mu_0 \int_0^{H_{max}} \left( \frac{\partial M}{\partial T} \right)_H dH$$

Here, $\mu_o$ is the magnetic permeability in vacuum, $M$ is the magnetization, $T$ is the temperature and $H$ is the applied magnetic field. The extracted $\Delta S_{M_\parallel}$ parallel to the $ab$-plane is shown in Fig. 2i. A broad asymmetric peak is observed with a maximum at 13 K for $\Delta \mu_0 H = 0.5$ T, which shifts to a higher temperature of 17 K for $\Delta \mu_0 H = 7$ T. For $\Delta \mu_0 H = 1, 2$ and 5 T, $\Delta S_M$ reaches values of −6.3, −11.4 and −18.3 J kg⁻¹ K⁻¹ respectively. The $M(\mu_0 H)$ curves measured perpendicular to the $ab$-plane, $M_\perp(\mu_0 H)$ and the corresponding $\Delta S_{M_\perp}$ are shown in Fig. S8 of the Supplementary Information. The easy-plane anisotropy of the system is apparent from the significantly smaller response of $M_\perp(\mu_0 H)$, and thereby $\Delta S_{M_\perp}$, compared to $M_\parallel(\mu_0 H)$ and $\Delta S_{M_\parallel}$.

An Arrott plot constructed from isothermal magnetization measurements and the scaling behavior of the peak magnetic entropy change $\Delta S_M^{Pk}$ with applied magnetic field, detailed in Section 6 of the Supplementary Information, shows that the phase transition can be classified as second order. Subsequent critical scaling analysis leads to the determination of the following critical exponents: $n = 0.489$, $\delta = 12.585$, $\beta = 0.134$ and $\gamma = 1.53$. These lie closest to the values expected for the 2D XY model with exponents $\beta = 1/8$, $\gamma = 7/4$ and $\delta = 15$[39]. A modified Arrott-Noakes plot and a universal curve (Supplementary Information Fig. S9) constructed using these exponents shows a set of nearly parallel curves which are linear around $T_c$ and the formation of two separate branches above and below $T_c$ respectively, validating this approach.

## Heat capacity and direct measurement of $\Delta T_{ad}$ in pulsed magnetic fields

Heat capacity measurements were performed on a sample of $Co_4(OH)_6(SO_4)_2[enH_2]$ consisting of an agglomerate of approximately aligned small crystals in applied magnetic fields of up to 10 T. As shown in Fig. 3a, with no applied magnetic field, a lambda-like peak is observed in the heat capacity at 10.2 K. With increasing magnetic field, the lambda-like peak decreases in magnitude and shifts towards higher temperatures.

Due to the lack of magnetostructural coupling as evidenced by the magnetic field-dependent PXRD measurements and the electronically insulating nature of $Co_4(OH)_6(SO_4)_2[enH_2]$, the total $\Delta S_M$, presented in Fig. 3b, is assumed to originate solely from its magnetic degrees of freedom. A single peak is observed in the $\Delta S_M(T)$ curve which shifts to higher temperatures with increasing magnetic field. For $\Delta \mu_o H = 1, 2$ and 5 T, $\Delta S_M$ reaches maximum values of −6.2, −8.7 and −15.3 J kg⁻¹ K⁻¹, respectively.

The polycrystalline nature of the sample leads to reduced $\Delta S_M$ values compared to the $\Delta S_{M_\parallel}$ values obtained from a single crystal in Fig. 2i. This stems from misalignment of the single crystalline domains that comprised the textured sample, combined with the strong magnetic anisotropy. Evidence for this claim is also found in $M(\mu_0 H)$ curves measured on part of the same sample used for the heat capacity measurements and the corresponding extracted $\Delta S_M$ (Supplementary Information Fig. S11). Fig. S11a shows mixed in-plane/out-of-plane behavior for the $M(\mu_0 H)$ curves and the corresponding $\Delta S_M$ shown in Fig. S11b is in excellent agreement with the values shown in Fig. 3b.

The heat capacity measurements also allow for the indirect determination of $\Delta T_{ad}$[40]. As shown in Fig. 3c, for $\Delta \mu_o H = 1, 2$ and 5 T, $\Delta T_{ad}$ reaches maximum values of 1.98, 3.22 and 5.88 K respectively.

In addition, $\Delta T_{ad}$ was directly measured on a single-crystalline sample in pulsed magnetic fields of up to 10 T at the Dresden High Magnetic Field Laboratory[41]. The directly determined $\Delta T_{ad}$ is shown alongside the indirectly determined $\Delta T_{ad}$ from the heat capacity measurements in Fig. 3c. Excellent agreement is observed between the two methods for $T > 25$ K. Upon approaching the critical temperature, a deviation between the two methods is observed. We argue that the higher $\Delta T_{ad}$ values derived from specific heat are more likely to reflect

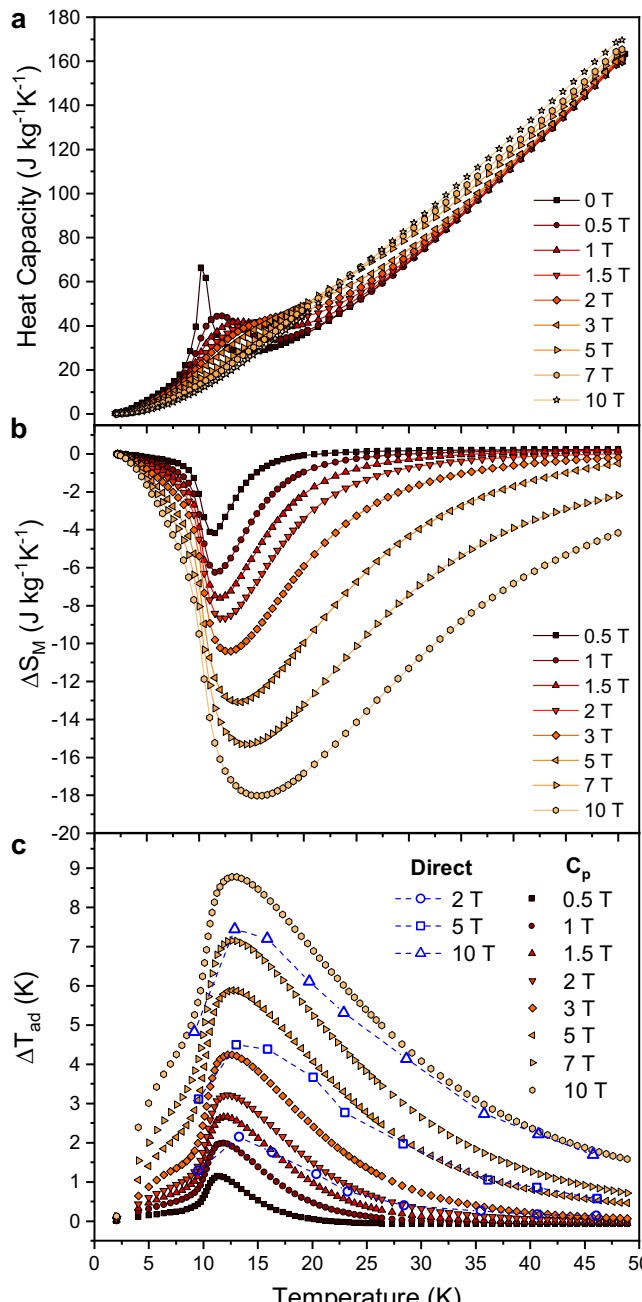

**Fig. 3 | Heat capacity, entropy change and adiabatic temperature change of Co₄(OH)₆(SO₄)₂[enH₂].** **a** Low temperature heat capacity plotted against temperature in applied magnetic fields ranging from 0 to 10 T. **b** Entropy change and **c** adiabatic temperature change plotted against temperature for magnetic field changes, $\Delta\mu_o H$, ranging from 0.5 to 10 T. In (**c**), the open symbols correspond to the direct measurement of the adiabatic temperature change in pulsed magnetic fields and the filled symbols correspond to values determined indirectly from the heat capacity measurement.

the intrinsic response of the material in the low-temperature regime. The sample preparation for the pulsed field measurements was challenging due to the ~100 μm thickness of the crystal. We expect that the silver epoxy used to attach the thermocouple to the sample and the GE varnish used to attach the sample to the holder will both act as heat sinks, inducing heat loss from the sample to the holder. This becomes more pronounced both when the absolute temperature difference between the sample and the holder is large and when the heat capacity of the sample is small at low temperatures, leading to a smaller than

expected $\Delta T_{ad}$. This notion is supported by the observation of hysteresis in a clockwise direction when the sample temperature is measured upon application and removal of the applied field, as shown in Fig. S12 of the Supplementary Information, and the fact that at higher temperatures, where $\Delta T_{ad}$ is smaller, no hysteresis is visible. We note that the values of $\Delta T_{ad}$ obtained from the specific heat measurements are still likely to be conservative, due to imperfect *ab*-plane alignment of the crystals that formed the polycrystalline sample.

## Discussion

The onset of magnetic order in good magnetocaloric materials often occurs with a character that resembles aspects of both first- and second-order phase transitions[42]. First-order transitions are advantageous in that the discontinuous change in magnetization at either $T_c$ or a critical applied field gives rise to a large peak in $\Delta S_M$. However, first-order transitions come with the disadvantage of hysteresis when cycling the temperature or applied field, and often a considerable discontinuous change in unit cell volume that can lead to mechanical fracture of the material. In this respect a second-order transition might be preferable, with no hysteresis and a continuous volume change, but at the expense of a smaller $\Delta S_M$.

Co₄(OH)₆(SO₄)₂[enH₂] exhibits the best of both worlds. The onset of magnetic order is second-order with minimal hysteresis and no detectable change in structure, but the sample is also easily magnetized in low fields with large $\Delta S_M$. The lack of hysteresis is particularly important because it allows for cyclic operation, which is not possible for many first-order materials whose properties are often reported for single measurement sweeps under laboratory conditions.

It is therefore instructive to examine the magnetic properties of Co₄(OH)₆(SO₄)₂[enH₂] in more detail in order to elucidate the origin of its favorable magnetocaloric performance. In particular, the small coercive field observed below $T_c$, which is significantly smaller than that of closely related compounds[38,43], might be attributed to several factors.

The pronounced easy-plane magnetic anisotropy and the XY character of the spins likely play an important role herein. Additionally, the derived Weiss temperatures from the in-plane and out-of-plane magnetization measurements indicate the presence of ferromagnetic (FM) and antiferromagnetic (AFM) interactions in-plane and out-of-plane, respectively. This is consistent with previously reported work on cobalt-based brucite-derived materials[43]. The relatively strong FM in-plane interactions are due to superexchange via Co-O-Co pathways that are close to 90°. However, the much weaker interlayer exchange can be AFM or FM depending on the interlayer distance, which in Co₄(OH)₆(SO₄)₂[enH₂] (~10.54 Å) is close to the critical distance of ~10 Å at which AFM superexchange becomes weaker than the FM dipolar interactions[43,44]. As a result, in brucite-derived materials with interlayer spacings larger than ~10 Å, a FM phase is observed[43,44]. As the interlayer spacing in Co₄(OH)₆(SO₄)₂[enH₂] is very close to this critical distance, the comparable strengths of the competing AFM and FM exchange interactions may be responsible for the small coercive field. Finally, the AC magnetic susceptibility measurements reveal that the magnetic ground state has a dynamic character. Frequency-dependent peaks are typically associated with glassy magnetic systems such as spin-glasses and superparamagnetic systems[45]. However, the single crystalline nature of the samples excludes a superparamagnetic origin and the lack of structural disorder in this compound as well as the other observed indicators of long-range ferromagnetic order indicate that these effects are not likely due to a spin-glass phase. Ferromagnetic materials such as ErAl₂, (Dy₀.₅Er₀.₅)Al₂, ErTi₂Ga₄ and DyTi₂Ga₄[46–49] display very similar AC responses to Co₄(OH)₆(SO₄)₂[enH₂], including the frequency-dependent peak observed in $\chi'(T)$ which sharply decreases below $T_c$, the observed splitting of the peaks in $\chi'(T)$ and $\chi''(T)$ and the opposite shift in temperature of the split $\chi'(T)$ peaks with increasing magnetic field.

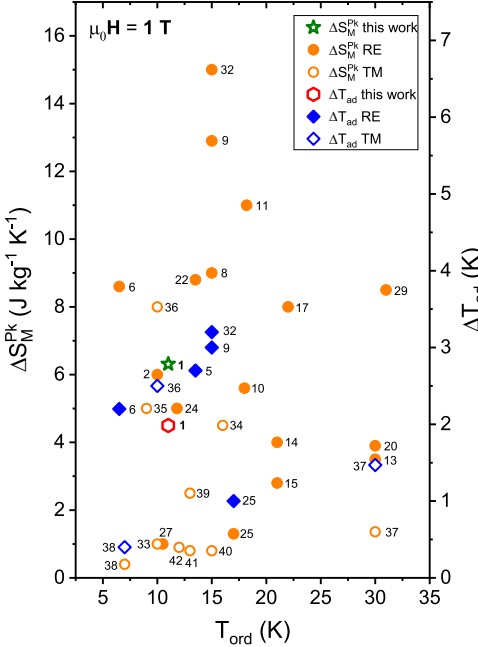

**Fig. 4 | Reported values of magnetic entropy and adiabatic temperature change of prospective magnetocaloric materials and Co$_4$(OH)$_6$(SO$_4$)$_2$[enH$_2$].** Values are plotted against their respective magnetic ordering temperatures ($T_{ord}$), for $\Delta\mu_0 H = 1$ T. The filled symbols indicate that the compound contains a rare-earth element and the open symbols indicate that the compound is based solely on transition-metal elements. The numeric labels refer to the compounds listed in Table S6 of the Supplementary Information. Values are estimated or taken from refs. [27,29–31,59–79]. Note that the peak values can be shifted by several Kelvin from $T_{ord}$.

These similarities suggest that a common process, namely dynamic domain wall movements[46–49], govern the behavior of $\chi'(T)$ and $\chi''(T)$ in this material. Materials exhibiting similar dynamic behavior show a trend of increasing coercivity with decreasing temperature starting from an almost non-hysteretic state at $T_c$[46–49].

Finally, we compare the performance of Co$_4$(OH)$_6$(SO$_4$)$_2$[enH$_2$] with other prospective magnetocaloric materials. Reported values of $\Delta S_M$ and $\Delta T_{ad}$ measured in a field of $\Delta\mu_0 H = 1$ T are shown in Fig. 4 for selected rare-earth based materials and for the best performing rare-earth free materials. Further comparisons for $\Delta\mu_0 H = 2$ T and 5 T can be found in Section 9 of the Supplementary Information. In most literature reports, $\Delta S_M$ and $\Delta T_{ad}$ values are quoted for high fields $\Delta\mu_0 H \geq$ 2 T, whereas the performance for $\Delta\mu_0 H \leq$ 2 T, provided by permanent magnets, can be more important for applications. Over the entire measured magnetic field range, the $\Delta S_M$ and $\Delta T_{ad}$ of Co$_4$(OH)$_6$(SO$_4$)$_2$[enH$_2$] are competitive with its rare-earth containing intermetallic counterparts and are maintained over a wide temperature range ($\Delta S_{FWHM}(10$ T$) \approx 25$ K). To the best of our knowledge the only other rare-earth free material with comparable performance is Co$_3$V$_2$O$_8$, which exhibits very similar peak values of $\Delta S_M$ and $\Delta T_{ad}$ (the latter not being directly measured but extracted from specific heat measurements) at a slightly lower temperature of 11 K[30]. However, there are other important aspects to consider when assessing the suitability of a material for future application in magnetocaloric cooling devices. First, the hydrothermal method used here to synthesize Co$_4$(OH)$_6$(SO$_4$)$_2$[enH$_2$] requires a temperature of only 170 °C and the product is obtained in a single step. Therefore, the synthesis can likely be scaled up readily. In contrast, Co$_3$V$_2$O$_8$ is prepared by a more labor-intensive solid-state method involving several heating steps up to 1100 °C with intermediate grinding between each. Second, although cobalt has been designated as a critical raw material,

Co$_4$(OH)$_6$(SO$_4$)$_2$[enH$_2$] contains approximately 30% less Co by weight for the same magnetocaloric performance in a 2 T magnetic field compared to Co$_3$V$_2$O$_8$. Moreover, Co$_4$(OH)$_6$(SO$_4$)$_2$[enH$_2$] is part of a family of layered metal hydroxides with great structural flexibility with respect to both molecular and atomic substitutions - thus we expect that the magnetocaloric properties can be optimized further and the material made more sustainable by the substitution of a more abundant transition metal such as Fe or Mn, and by fine tuning of the interlayer gap. Third, high thermal conductivity is an important secondary functional property for magnetocaloric materials. Co$_3$V$_2$O$_8$ exhibits a thermal conductivity of ~15 W m$^{-1}$ K$^{-1}$ at 20 K with almost no anisotropy[50]. Although the thermal conductivity of Co$_4$(OH)$_6$(SO$_4$)$_2$[enH$_2$] has not yet been measured, other coordination polymers can achieve substantially larger values, for example the layered material Gd(HCOO)(C$_2$O$_4$)]$_n$ exhibits a value of ~45 W m$^{-1}$ K$^{-1}$ at 8 K[51]. The likely anisotropic thermal conductivity of Co$_4$(OH)$_6$(SO$_4$)$_2$[enH$_2$] might be a further advantage, minimizing unwanted heat transfer to the surroundings in certain directions.

In summary, we report the synthesis of a novel layered hybrid coordination polymer Co$_4$(OH)$_6$(SO$_4$)$_2$[enH$_2$] and a thorough investigation of its structural and magnetocaloric properties. For small magnetic field changes that permanent magnets can supply, $\Delta S_M$ and $\Delta T_{ad}$ reach values that are exceptional for non-rare-earth based materials and are competitive with good rare-earth containing alloys for magnetic cooling around the hydrogen liquefaction temperature range. Moreover, Co$_4$(OH)$_6$(SO$_4$)$_2$[enH$_2$] can be synthesized by a facile hydrothermal method, requiring a temperature of only 170 °C. We show that the excellent magnetocaloric properties can be attributed to a second-order ferromagnetic phase transition with 10 K $< T_c <$ 15 K. The layered character of the crystal structure likely plays a central role herein, leading to easy-plane magnetic anisotropy with XY spins that can easily be aligned in small magnetic fields, with a corresponding large magnetic entropy change and minimal hysteresis on removing or reversing the field.

We hope that this work will stimulate further research into similar materials as the brucite-type structure of Co$_4$(OH)$_6$(SO$_4$)$_2$[enH$_2$] is highly flexible with respect to possible substitutions of both the transition metal and the ions in the interlayer space. We expect that there is still room for improving the magnetocaloric properties, for instance by incorporating transition metals with larger magnetic moments such as Fe$^{2+}$ ($S = 2$) or Mn$^{2+}$ ($S = 5/2$).

## Methods

### Synthesis

All reagents and solvents except for deionized water were obtained from commercial vendors and used as received. The starting materials, CoSO$_4$·7H$_2$O (4 mmol, 1.127 g, Sigma-Aldrich, ≥99%), ethylenediamine (1.66 mmol, 0.11 mL, Fluka analytical, ≥99.5 %), H$_2$SO$_4$ (0.5 mmol, 0.027 mL, Merck, 95-97%) and deionized water (4 mmol, 0.075 mL) were added to a 25 mL Teflon insert. The Teflon insert was placed in a stainless steel autoclave which was sealed and subsequently heated in a Pol-Eko oven to 170 °C over 5 h and kept at that temperature for 5 days, after which it was cooled down to room temperature over 10 hours. The resulting product was washed with deionized water and ethanol.

### Exfoliation

Nanosheets of Co$_4$(OH)$_6$(SO$_4$)$_2$[enH$_2$] were obtained by the addition of single crystals of Co$_4$(OH)$_6$(SO$_4$)$_2$[enH$_2$] (~30 mg) to a glass vial containing 15 mL 96 % ethanol and subsequent sonication for 60 min.

### SCXRD

Single crystal X-ray diffraction measurements were performed using a Bruker D8 Venture diffractometer equipped with a Photon 100 CMOS detector operating with Mo Kα radiation at 107 K. Two twin

components were identified using the CELL_NOW[52] routine (180° rotation around the [110] axis). Data integration was performed using the SAINT[53] routine within the Bruker Apex II software package[54] and multi-scan absorption correction was performed using TWINABS[55]. The subsequent twinned structure refinement was performed using SHELXL[56].

## FT-IR
Fourier transform-infrared spectroscopy measurements were performed directly on ground powder samples using a PerkinElmer Spectrum 400 FT-IR spectrophotometer from 4000 to 650 cm$^{-1}$ equipped with an attenuated total reflection sample stage.

## PXRD
The temperature and magnetic field dependent powder x-ray diffraction measurements were performed on a custom-built diffractometer equipped with a MYTHEN detector and a superconducting magnet in transmission Debye-Scherrer geometry using Mo x-ray radiation ($\lambda_1 = 0.70932$ Å, $\lambda_2 = 0.7134$ Å). Powdered samples were deposited alongside silicon powder (for correcting geometric errors) on a graphite sample support[57]. Rietveld refinements of the measured powder patterns were performed using the Fullprof software suite[58].

## AFM
Samples were prepared by the submersion of silicon substrates, which were cleaned prior by sonication in 96 % ethanol, deionized water and acetone baths, in a glass vial containing the exfoliated $Co_4(OH)_6(SO_4)_2[enH_2]$ nanosheets. The substrates were subsequently dried in air. Atomic force microscopy images were collected on a Bruker Dimension Icon microscope in tapping mode.

## (S)TEM
Samples were prepared by submersion of a lacey carbon film covered copper grid into a glass vial containing the exfoliated $Co_4(OH)_6(SO_4)_2[enH_2]$ nanosheets. Scanning transmission electron microscopy images were collected using a Thermo Fisher Scientific Themis Z S/TEM operating at 300 kV using a probe current of less than 1 pA. A liquid nitrogen cooled sample stage was used during the measurements. We note that in TEM mode the samples were damaged after a few seconds of beam exposure.

## DC magnetization
Temperature and magnetic field dependent DC magnetization measurements were performed using a Quantum Design MPMS XL-7 T SQUID magnetometer. Magnetic hysteresis loops were acquired by cycling the field from $+H_{max}$ to $-H_{max}$ and then back from $-H_{max}$ to $+H_{max}$ The field dependent magnetization measurements used for the determination of the magnetic entropy change were performed on warming by measuring from $+H_{max}$ to 0 T.

## AC magnetic susceptibility
The real ($\chi'$) and imaginary ($\chi''$) components of the frequency and temperature dependent AC magnetic susceptibility measurements were determined using the ACMS II option for the Quantum Design PPMS system in zero-field-cooled conditions using various DC bias field strengths in the range of 0-500 mT and an AC driving field of 0.43 mT.

## Heat capacity
Heat capacity measurements were performed on cooling using the Heat Capacity option for the Quantum Design PPMS system. Magnetic field-dependent addenda measurements were obtained in applied magnetic fields of 0, 2, 5 and 10 T. A sample comprised of an agglomerate of small crystals with similar but not identical alignments (3.585 mg) was thermally anchored to the sample stage using Apiezon-N grease. Measurements were performed using the high-vacuum option of the PPMS and the pressure during measurements was typically 1.23 mPa. The adiabatic temperature change is determined as $\Delta T_{ad} = T(S)_H - T(S)_0$ and the isothermal entropy change is determined as $\Delta S_M = S(T)_H - S(T)_0$. $T(S)$ and $S(T)$ can be obtained by interpolating the experimental entropy curves acquired under the respective magnetic field.

## Adiabatic temperature change
The adiabatic temperature change was measured directly at the Dresden High Magnetic Field Laboratory in applied magnetic fields of 2, 5 and 10 T using a pulsed field magnet. Two thin single crystals with a thickness of roughly 100 μm were glued together using silver epoxy. In between the crystals, a type E thermocouple with a wire thickness of 25 μm was placed. The sample was fixed to the sample holder with GE varnish. The magnetic field was aligned along the *ab*-plane. The time required to reach the maximum field was 19 ms for all experiments.

## Data availability
The data generated in this study have been deposited in the Dataverse.nl database under accession code https://doi.org/10.34894/ZPLHDY. The CIF data for the 107 K crystal structure have been deposited in the Cambridge Crystallographic Data Centre (CCDC) under deposition number 2383963.

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

## Acknowledgements

The authors thank J. Baas for technical support. J.J.B.L. was supported by the research program "Skyrmionics: towards new magnetic skyrmions and topological memory" of the Netherlands Organization for Scientific Research (NWO, Project No. 16SKYR03). B.B., D.K., and O.G. gratefully acknowledge the funding by the German Research Foundation (DFG) within the Collaborative Research Centre/Transregio (CRC/TRR) 270 (Project-ID 405553726, project B01 and B04) and DFG grant No. INST 163/442-1 FUGG. T.G. was supported by the HLD at HZDR (member of the European Magnetic Field Laboratory (EMFL)) and by the Clean Hydrogen Partnership and its members within the framework of the project HyLICAL (Grant No. 101101461).

## Author contributions

G.R.B. conceived and supervised the project. J.J.B.L. performed the synthesis and the SCXRD, AFM, FT-IR and DC/AC magnetic susceptibility experiments. T.G. performed the adiabatic temperature change measurements. B.B. performed the heat capacity measurements and additional magnetization measurements. M.A. performed the (S)TEM measurements. D.K. performed the PXRD measurements. J.J.B.L. wrote the manuscript with contributions from G.R.B., B.B., T.G., and O.G.

## Competing interests

The authors declare no competing interests.
