## [Peer Review File · Nature Communications]

Reviewers' Comments:

Reviewer #1:

Remarks to the Author:

Authors reported a study on a Rare-Earth-Free Layered Coordination Polymer which can be suitable for operation at Liquid Hydrogen Temperatures. They found that, for small fields that permanent magnets can supply, ΔS and ΔT_{ad} of these materials can reach giant MCE values that are competitive with or even surpass the current best rare-earth containing alloys for magnetic cooling. They claim that they discover one material which shows the largest magnetocaloric response for an organic-inorganic hybrid material in this temperature range. The study is interesting for a wide range of readers and the results are fascinating. The proposed materials and the related experimental results are encouraging and have been well explained. So, it therefore deserves publication.

However, I have the following concerns,

It is better that the authors provide the temperature dependence of lattice parameters because they already measured the x-ray diffraction patterns as a function of temperature (shown in Figure S3). Some readers may be interested to know if there exists one anomaly in thermal expansion (the potential presence of magnetoelastic coupling). Similarly, they may also provide the field dependence of lattice parameters at $T=15$ K based on their Figure S4 data.

The authors need add more discussions on their results related to recent progress in this field. For an example, there are a few systems recently being reported which can be used in the similar temperature regions with excellent performance such as RMn_2X_2 -related system (TbMn₂Si₂-based, NdMn₂Si₂-based materials etc.).

Once this is done, the impact of the work will be visible. So, I recommend 'minor Revision' for publication.

Reviewer #2:

Remarks to the Author:

In this manuscript, Levinsky and his coworkers report magnetocaloric effect of single crystal of organic and inorganic hybrid compound $Co_4(OH)_6(SO_4)_2[enH_2]$, by means of the magnetization and specific heat measurements. This work might be important to provide a rare-earth-free magnetic cooling material near the hydrogen liquefaction temperature range, due to the recent society demand of hydrogen as an energy storage medium. The authors argued that this material shows the largest magnetic entropy change (ΔS) near the hydrogen liquefaction temperature in all rare-earth-free systems. If it was scientifically convincing, this work would be important work. However, there are serious discrepancies in the ΔS values given by magnetization and specific heat results. In Fig. 2(i), they showed the temperature dependence of ΔS

with the maximum of ~ -6 J/kg K at 15 K and 1 T, which was derived from the magnetization data in Fig. 2(g) by using the Maxwell relation. I think that the magnetization data are convincing. The Δ_S value in Fig. 2(i) is reasonable in a transition metal ferromagnet with 3 μ_B saturation moments. On the other hand, the authors provide the completely different Δ_S curves in Fig. 3(d). The discrepancy is seriously problematic. The Δ_S (Fig. 3e) shows the maximum at 30 K and 1 T, the value was -12 J/kg K. Obviously, the discrepancy can be understood to be caused by the additional large broad peak contribution around 35 K. The authors are arguing that the broad peak in specific heat and Δ_S (Δ_T) can be magnetic in origin due to magnetic short-range order without any ferromagnetic components. However, if Δ_S was magnetic in origin, Δ_S must obey the Maxwell relation, because the net magnetization (ferromagnetic component, M) is a conjugate physical quantity against a uniform magnetic field. Since this compound shows only $M=0.5$ μ_B/Co at 30 K and 1 T, such the large Δ_S of -12 J/kg K cannot be magnetic in origin.

In conclusion, due to the serious problematic discrepancy between two measurements, the data provided in this paper (especially for specific heat data) are not scientifically valid. I therefore think that this paper is not suitable for publication in any form.

I strongly recommend of the authors to revisit the specific heat measurements with better single crystals with appropriate mass, and also to measure adiabatic temperature change (Δ_T) by a direct measurement without any ambiguity.

We are grateful to the Reviewers for their rigorous assessment of our manuscript and for pointing out some problems that we had failed to appreciate properly. In order to address these issues we needed to expand our team, perform some new experiments on larger samples and reassess some of the data, particularly the specific heat capacity. We now believe that we have a robust manuscript where the conclusions are fully supported by the data. The magnetocaloric performance parameters that we now report (ΔS , ΔT) are somewhat lower than in the previous version of our manuscript, but they are still competitive with rare-earth based magnetocaloric materials operating near the hydrogen liquefaction temperature and to the best of our knowledge are still the highest among all organic-inorganic hybrid materials in this temperature range. Therefore we believe that the importance of our work is still as relevant as earlier.

We outline below in red text our responses to the points raised by the Reviewers.

Reviewer #1 (Remarks to the Author):

Authors reported a study on a Rare-Earth-Free Layered Coordination Polymer which can be suitable for operation at Liquid Hydrogen Temperatures. They found that, for small fields that permanent magnets can supply, ΔS and ΔT_{ad} of these materials can reach giant MCE values that are competitive with or even surpass the current best rare-earth containing alloys for magnetic cooling. They claim that they discover one material which shows the largest magnetocaloric response for an organic-inorganic hybrid material in this temperature range. The study is interesting for a wide range of readers and the results are fascinating. The proposed materials and the related experimental results are encouraging and have been well explained. So, it therefore deserves publication.

We are pleased that the Reviewer appreciates the importance of the study, which we believe is still valid as to the best of our knowledge there have been in the meantime no studies published on the magnetocaloric properties of any materials similar to $\text{Co}_4(\text{OH})_6(\text{SO}_4)_2[\text{enH}_2]$.

However, I have the following concerns,

It is better that the authors provide the temperature dependence of lattice parameters because they already measured the x-ray diffraction patterns as a function of temperature (shown in Figure S3). Some readers may be interested to know if there exists one anomaly in thermal expansion (the potential presence of magnetoelastic coupling). Similarly, they may also provide the field dependence of lattice parameters at $T=15$ K based on their Figure S4 data.

We have added plots of the temperature and magnetic field dependence of the lattice parameters in Supplementary Figs. S5 and S6. No obvious anomalies are apparent, thus there does not seem to be any appreciable magnetoelastic coupling.

The authors need add more discussions on their results related to recent progress in this field. For an example, there are a few systems recently being reported which can be used in the similar temperature regions with excellent performance such as RMn_2X_2 -related system (TbMn_2Si_2 -based, NdMn_2Si_2 -based materials etc.).

We have added extra references 13-16 in the introduction, which include some studies on $\text{NdMn}_{2-x}\text{M}_x\text{Si}_2$ compounds.

Once this is done, the impact of the work will be visible. So, I recommend 'minor Revision' for publication.

Reviewer #2 (Remarks to the Author):

In this manuscript, Levinsky and his coworkers report magnetocaloric effect of single crystal of organic and inorganic hybrid compound $\text{Co}_4(\text{OH})_6(\text{SO}_4)_2[\text{enH}_2]$, by means of the magnetization and specific heat measurements. This work might be important to provide a rare-earth-free magnetic cooling material near the hydrogen liquefaction temperature range, due to the recent society demand of hydrogen as an energy storage medium. The authors argued that this material shows the largest magnetic entropy change (ΔS) near the hydrogen liquefaction temperature in all rare-earth-free systems. If it was scientifically convincing, this work would be important work. However, there are serious discrepancies in the ΔS values given by magnetization and specific heat results. In Fig. 2(i), they showed the temperature dependence of ΔS with the maximum of ~ -6 J/kg K at 15 K and 1 T, which was derived from the magnetization data in Fig. 2(g) by using the Maxwell relation. I think that the magnetization data are convincing. The ΔS value in Fig. 2(i) is reasonable in a transition metal ferromagnet with 3 μB saturation moments. On the other hand, the authors provide the completely different ΔS curves in Fig. 3(d). The discrepancy is seriously problematic. The ΔS (Fig. 3e) shows the maximum at 30 K and 1 T, the value was -12 J/kg K. Obviously, the discrepancy can be understood to be caused by the additional large broad peak contribution around 35 K. The authors are arguing that the broad peak in specific heat and ΔS (ΔT) can be magnetic in origin due to magnetic short-range order without any ferromagnetic components. However, if ΔS was magnetic in origin, ΔS must obey the Maxwell relation, because the net magnetization (ferromagnetic component, M) is a conjugate physical quantity against a uniform magnetic field. Since this compound shows only $M=0.5$ $\mu\text{B}/\text{Co}$ at 30 K and 1 T, such the large ΔS of -12 J/kg K cannot be magnetic in origin.

In conclusion, due to the serious problematic discrepancy between two measurements, the data provided in this paper (especially for specific heat data) are not scientifically valid. I therefore think that this paper is not suitable for publication in any form.

I strongly recommend of the authors to revisit the specific heat measurements with better single crystals with appropriate mass, and also to measure adiabatic temperature change (ΔT) by a direct measurement without any ambiguity.

We are very grateful to the Reviewer for pointing out this problem in the previous version of our manuscript. We agree that the ΔS values obtained from specific heat and magnetization measurements were not fully consistent, caused indeed by the broad peak-like feature in the specific heat curve centred at 35 K. Although we had performed repeated measurements on single crystals before submission of the original version of our manuscript, which gave reproducible results, we have since carried out specific heat measurements on much larger samples comprised of agglomerates of approximately aligned crystals in a different lab. These new measurements no longer exhibit the broad feature at 35 K and the extracted values of ΔS agree well with those obtained from the magnetization measurements (see Figs. 2i and 3b of the revised manuscript). Therefore we can rule out any significant influence of short-range magnetic order on ΔS . We are still unsure of what caused the apparent measurement artefact in our previous specific heat data, but it might indeed have been due to a too small sample size.

We have also followed the recommendation to perform direct measurements of ΔT . This is best done using a pulsed magnetic field to ensure a rapid change in magnetization with time, and we followed a protocol previously developed at the Dresden High Magnetic Field Laboratory, using pulsed fields up to 10 T. The results are shown in Fig. 3c, where they are compared with ΔT extracted from the new specific heat measurements. The agreement is rather good, and we explain that the slightly lower values of ΔT obtained from the direct measurements can probably be attributed to a degree of heat loss from the sample to the holder, as evidenced by hysteresis in temperature versus applied field curves (see Fig. S12). This is unavoidable in these rather difficult direct ΔT measurements. We believe that the ΔT values extracted from the specific heat measurements are also on the conservative side due to imperfect alignment of the cluster of crystals used (as evidenced by a lower saturation magnetization than obtained for a single crystal – see Fig. S11). These issues are outlined in the completely rewritten Sections 2C and 3.

The conclusion is that ΔS and ΔT for our material are lower than reported in the previous version of our manuscript (see updated Figs. 4 and S13), but for the reasons above they are probably somewhat underestimated. Nevertheless, they are still competitive with rare-earth-based magnetocaloric materials in this temperature range and to the best of our knowledge there have been no reports in the intervening time of new transition-metal-based systems with better performance. We thus believe that the importance of our work is not diminished, showing the great promise of layered organic-inorganic hybrid materials for low-temperature magnetocaloric applications.

Reviewers' Comments:

Reviewer #1:

Remarks to the Author:

The authors have answered the questions which I raised in the previous version. I can accept this paper to be published.

Reviewer #2:

Remarks to the Author:

The authors have revised the manuscript titled "Giant Magnetocaloric Effect in a Rare-Earth-Free Layered Coordination Polymer at Liquid Hydrogen Temperatures". As mentioned in my previous review, the paper would be worth being published in this journal only if the extremely large magnetocaloric value of Δ_{SM} of ~ 12 J/K kg at 35 K in the rare earth free compound was true. I specified that there is the serious discrepancy the Δ_{SM} values given by magnetization and specific heat results and that the large Δ_{SM} value can be caused by experimental error. I therefore concluded that "Due to the serious problematic discrepancy between two measurements, the data provided in this paper (especially for specific heat data) are not scientifically valid."

In the revised paper, the authors recognized that the large Δ_{SM} value (~ 12 J/K kg at 35 K) was not true but caused by the experimental error. Then, they reported the smaller Δ_{SM} value of ~ 6 J/K kg at 15 K. They spent large efforts to remeasure the specific heat data with larger sample with appropriate mass and added the adiabatic temperature change data by using the direct measurement, which are consistent with each other. I think that the experimental data in the revised version are scientifically valid.

However, the Δ_{SM} value reported in the revised paper is normal value even for non-rare-earth compounds. (for example, in $\text{Co}_3\text{V}_2\text{O}_8$, it shows ~ 9 J/K kg around 10 K at 1 T.) I think that the paper is now lacking novelty.

In conclusion, I do not recommend that this paper is suitable for publication in this journal, and it should be published in a specialized journal.

Response to Reviewers

We appreciate the attention given to our manuscript by the two Reviewers. We are pleased that Reviewer #1 now considers our work suitable for publication, and that Reviewer #2 is now satisfied with the technical correctness of our work. However, we respectfully disagree with the main criticism of Reviewer #2 that “the ΔS_M value reported in the revised paper is a normal value even for non-rare-earth compounds” and thus that the manuscript “is now lacking novelty”. Even though the values of magnetic entropy change in different applied fields are indeed lower than in our initial submission, we strongly disagree that they are “normal”.

The Reviewer backs up this statement by referring to $\text{Co}_3\text{V}_2\text{O}_8$ as an example, the properties of which are reported in a paper published in January 2024 (10.1103/PhysRevB.109.014418). We are grateful to the Reviewer for bringing this work to our attention; unfortunately we had missed it, perhaps because the magnetocaloric properties of $\text{Co}_3\text{V}_2\text{O}_8$ were rather surprisingly not presented as the sole focus of that paper. Indeed, $\text{Co}_3\text{V}_2\text{O}_8$ exhibits very similar values of ΔS_M and ΔT (although the latter was not measured directly) to our compound $\text{Co}_4(\text{OH})_6(\text{SO}_4)_2[\text{enH}_2]$. We have since performed a thorough literature search to check whether we have missed any other non-rare-earth based magnetocaloric materials with similar or better performance in the hydrogen liquefaction temperature range, but to the best of our knowledge this is not the case. In Figure 1 below are two plots of ΔS_M for the best non-rare-earth based compounds in this temperature range, measured in applied magnetic fields of 1 T and 2 T. It is clear that $\text{Co}_4(\text{OH})_6(\text{SO}_4)_2[\text{enH}_2]$ and $\text{Co}_3\text{V}_2\text{O}_8$ are well ahead of the rest, thus these are clearly not “normal” values.

Furthermore, we argue that the following important aspects make our study unique and novel.

1. We have performed direct ΔT measurements using pulsed magnetic fields to confirm the values extracted “indirectly” from specific heat capacity data, which was not done for $\text{Co}_3\text{V}_2\text{O}_8$.
2. $\text{Co}_4(\text{OH})_6(\text{SO}_4)_2[\text{enH}_2]$ is synthesised at 170 °C using a single-step hydrothermal method that can easily be scaled up, very much preferable to the 1100 °C needed for the solid-state synthesis of $\text{Co}_3\text{V}_2\text{O}_8$ with energy efficiency in mind.
3. $\text{Co}_4(\text{OH})_6(\text{SO}_4)_2[\text{enH}_2]$ contains approximately 30% lower weight% of the “critical” element Co for the same magnetocaloric performance in a 2 T field compared to $\text{Co}_3\text{V}_2\text{O}_8$. Moreover, $\text{Co}_4(\text{OH})_6(\text{SO}_4)_2[\text{enH}_2]$ is part of a family of layered metal hydroxides with great structural flexibility with respect to both molecular and atomic substitutions. We thus expect that the magnetocaloric properties can be improved further and

Figure 1: Comparison of maximum ΔS_M values and their corresponding temperatures in applied magnetic fields of 1 T (left) and 2 T (right) for the best performing non rare-earth based magnetocaloric materials reported in the literature.

the material made more “sustainable” by the substitution of a more abundant transition metal cation (Fe^{2+} or Mn^{2+} , both likely having a larger magnetisation than Co^{2+}) and / or by tuning the inter-layer gap. We expect that $\text{Co}_4(\text{OH})_6(\text{SO}_4)_2[\text{enH}_2]$ will open up a new field of magnetocalorics research focusing on layered, transition-metal-based coordination polymers.

4. High thermal conductivity is an important secondary functional property of magnetocaloric materials. $\text{Co}_3\text{V}_2\text{O}_8$ is a well-studied material with a thermal conductivity of approximately 15 W/(m.K) at 20 K with almost no anisotropy ([10.1063/1.4953790](https://doi.org/10.1063/1.4953790)). Although we have not measured the thermal conductivity of $\text{Co}_4(\text{OH})_6(\text{SO}_4)_2[\text{enH}_2]$, other magnetocaloric coordination polymers are known to achieve substantially larger values, for example $\text{Gd}(\text{HCOO})(\text{C}_2\text{O}_4)]_n$ exhibits a value of 45 W/(m.K) at 8 K ([10.1021/acs.inorgchem.1c01152](https://doi.org/10.1021/acs.inorgchem.1c01152)). The likely anisotropic thermal conductivity of our compound could be a further advantage for device construction, minimizing unwanted heat transfer to the surroundings in certain directions.

We therefore believe that the unique properties of our novel compound, with the prospect of further enhanced properties for future materials derived from it or based on its structure type, are of interest to the broad readership of Nature Communications.

We have made changes to the manuscript to better emphasise the most important features of $\text{Co}_4(\text{OH})_6(\text{SO}_4)_2[\text{enH}_2]$ outlined above. In the Introduction we have extended the discussion of other transition-metal-based magnetocaloric materials with crystal structures that can be considered layered in nature, in order to better put our work in context. We have updated Figure 4, Figure S13 and Table S6, to include data on other transition-metal-based magnetocaloric materials in the literature (as also shown in the figure above). Corresponding references have been added. Text has been added on page 10-12 to discuss some desirable properties of $\text{Co}_4(\text{OH})_6(\text{SO}_4)_2[\text{enH}_2]$ that go beyond the high values of ΔS_M and ΔT .